# Evolutionary convergence of a neural mechanism in the cavefish lateral line system

Elias T Lunsford[1], Alexandra Paz[2], Alex C Keene[3], James C Liao[1]*

[1]Department of Biology, University of Florida, Saint Augustine, United States; [2]Department of Biological Sciences, Florida Atlantic University, Jupiter, United States; [3]Texas A&M University, College Station, United States

**Abstract** Animals can evolve dramatic sensory functions in response to environmental constraints, but little is known about the neural mechanisms underlying these changes. The Mexican tetra, *Astyanax mexicanus*, is a leading model to study genetic, behavioral, and physiological evolution by comparing eyed surface populations and blind cave populations. We compared neurophysiological responses of posterior lateral line afferent neurons and motor neurons across *A. mexicanus* populations to reveal how shifts in sensory function may shape behavioral diversity. These studies indicate differences in intrinsic afferent signaling and gain control across populations. Elevated endogenous afferent activity identified a lower response threshold in the lateral line of blind cavefish relative to surface fish leading to increased evoked potentials during hair cell deflection in cavefish. We next measured the effect of inhibitory corollary discharges from hindbrain efferent neurons onto afferents during locomotion. We discovered that three independently derived cavefish populations have evolved persistent afferent activity during locomotion, suggesting for the first time that partial loss of function in the efferent system can be an evolutionary mechanism for neural adaptation of a vertebrate sensory system.

*For correspondence:
jliao@whitney.ufl.edu

Competing interest: The authors declare that no competing interests exist.

## Editor's evaluation

While most cavefish lack vision, their blindness is compensated by enhanced sensitivity to water flows. Through a powerful comparative analysis of neural circuits, the present work sheds light on how sensory mechanisms have convergently evolved in multiple populations of fish that have independently colonized different cave environments.

## Introduction

Our understanding of the sensory systems and behavior of animals is challenging to contextualize within the framework of evolution. Anatomical comparisons between species have allowed us to infer sensory capabilities, but this approach cannot directly reveal neural function. Discovery of neural mechanisms that underlie behavior are often constrained to a limited number of model species (*Jourjine and Hoekstra, 2021*). Like morphology, neural circuits can adapt to the environment. Of these, many circuits are sensory and regulate essential behaviors such as foraging, navigation, and escapes (*Blin et al., 2018*; *Hoke et al., 2012*; *Hüppop, 1987*; *Paz et al., 2020*).

The Mexican blind cavefish, *Astyanax mexicanus*, is a powerful model to understand the evolution of physiological and molecular traits that contribute to behaviors such as sleep (*Duboué et al., 2011*; *Jaggard et al., 2018*) and prey capture (*Lloyd et al., 2018*; *Yoshizawa et al., 2014*). *A. mexicanus* exists in two morphs: (1) eyed surface-dwelling populations and (2) blind cave populations. There are

at least 30 independently evolved cavefish populations in the caves of the Sierra de El Abra region of Northeast Mexico (*Espinasa et al., 2018*; *Herman et al., 2018*; *McGaugh et al., 2020*; *Mitchell et al., 1997*). *A. mexicanus* populations are interfertile, and this attribute has allowed investigators to demonstrate independent convergence of numerous behavioral, developmental, and physiological traits (*Chin et al., 2018*; *Kowalko, 2020*; *Riddle et al., 2018*; *Stockdale et al., 2018*; *Varatharasan et al., 2009*). Our goal is to apply a neurophysiological approach across multiple *A. mexicanus* populations to examine the functional evolution of the lateral line sensory system.

The mechanoreceptive hair cells of the lateral line system detects fluid motion relative to the body and play an important role in essential behaviors (*McHenry et al., 2009*; *Mekdara et al., 2018*; *Olszewski et al., 2012*; *Oteiza et al., 2017*; *Stewart et al., 2013*). Cavefish have evolved anatomical enhancements of the lateral line, presumably to compensate for the loss of vision (*Kowalko, 2020*; *McGaugh et al., 2020*; *Teyke, 1990*; *Yoshizawa et al., 2012*). These anatomical alterations have been linked to substantial changes in behavior (*Lloyd et al., 2018*; *Yoshizawa et al., 2010*). However, almost nothing is known about physiological changes that accompany evolution, despite the fact that the response of peripheral senses to environmental change has been well documented (*Kelley et al., 2018*; *McBride, 2007*).

Endogenous depolarizations within sensory cells are transmitted to afferent neurons (hereafter 'afferents') as spontaneous action potentials (hereafter 'spikes'), and are essential for maintaining a state of responsiveness and sensitivity (*Kiang, 1965*; *Manley and Robertson, 1976*). For example, the spontaneous spikes from auditory hair cells have been shown to enable precise encoding of much higher frequencies (*Köppl, 1997*). Here, we suggest a similar underlying mechanism, though the correlation of spontaneous activity to sensitivity in the lateral line awaits empirical evidence. In the lateral line, spontaneous depolarizing currents within the hair cell maintain a resting potential within the critical activation range of channels. This range is required to ensure transmitter release; thus these currents decrease the detection threshold of the system (*Trapani and Nicolson, 2011*). Spontaneous afferent activity is an established and reliable target for probing the neurophysiological basis of sensitivity across taxa (*Hedwig, 2006*; *Krasne and Bryan, 1973*; *Mohr et al., 2003*). Here, we use spontaneous afferent activity as an entry point into understanding the neural mechanism underlying lateral line function in cavefish.

Another important mechanism that determines lateral line sensitivity is an inhibitory feedback effect of the efferent system during swimming. Feedback mechanisms in general sculpt sensory systems in important ways; for example, by changing detection thresholds by altering the transmission frequency of afferent spikes (*Crapse and Sommer, 2008*; *Straka et al., 2018*). The efferent system of hair cells in particular has repeatedly evolved to modulate sensory processing (*Köppl, 2011*). More specifically, hindbrain efferent neurons (hereafter 'efferents') issue predictive signals that transmit in parallel to locomotor commands, termed corollary discharge (CD). CDs inhibit afferent activity to mitigate sensor fatigue that can result from self-generated stimuli (*Russell and Roberts, 1972*). CD is an important mechanism for sensitivity enhancement but has rarely been implicated for ecologically relevant behaviors. For example, active-flow sensing by cavefish depends on detecting reafferent signals while swimming (*Tan et al., 2011*). Increased reliance on self-generated fluid motion (*Odstrcil et al., 2022*; *Patton et al., 2010*; *Teyke, 1985*) is divergent from our current understanding of the CD's role in predictive motor signaling in the lateral line (*Lunsford et al., 2019*; *Pichler and Lagnado, 2020*).

For the first time, we identify a neurophysiological mechanism that has convergently evolved across *A. mexicanus* populations to increase hair cell sensitivity after eye loss. By investigating how differences in afferent and efferent signaling contribute to sensory enhancement in a comparative model, we provide insight into a potentially ubiquitous mechanism for sensory evolution.

## Results

Neuromasts of surface fish and Pachón cavefish larvae (6 days post fertilization; dpf) were labeled via 2-[4-(dimethylamino)styryl]-1-ethylpyridinium iodide (DASPEI) staining and subsequently imaged (*Figure 1B, C*). Anterior lateral line (ALL) neuromasts had previously been shown to differ in quantities and morphology as early as 2 months post fertilization between surface and cavefish (*Yoshizawa et al., 2010*). Here, we show that Pachón cavefish exhibit this significant increase in anterior neuromast quantity as early as 6 dpf ($p < 0.01$, $t = 3.168$, df = 29; *Figure 1D*). In contrast, both populations exhibit a similar number of posterior lateral line neuromasts ($p = 0.77$, $U = 111.5$; *Figure 1E*).

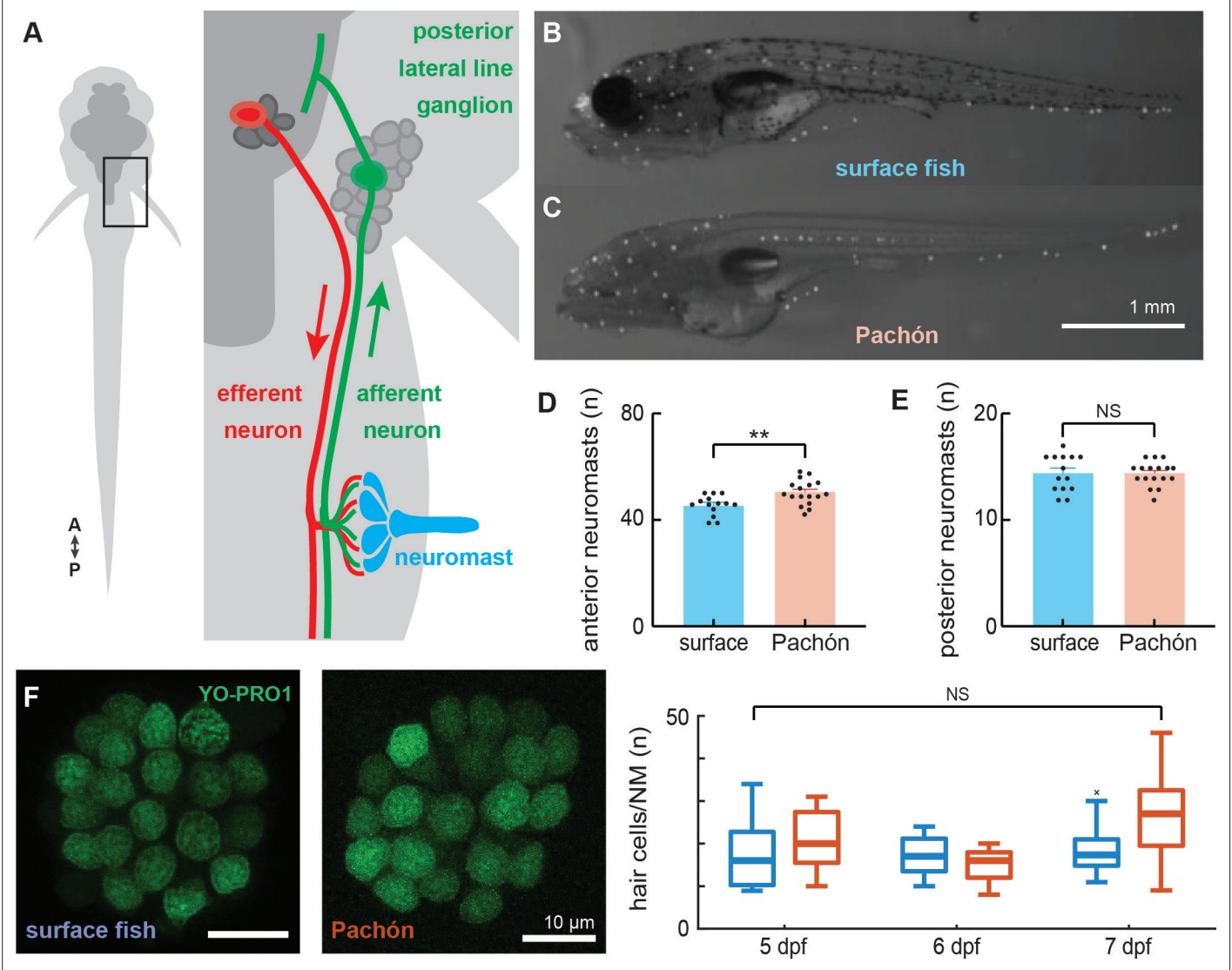

**Figure 1.** Surface fish and cavefish lateral lines are anatomically similar in early development. Illustration (not to scale) of the canonical circuit of lateral line function depicting the neuromast (blue) comprised of mechanoreceptive hair cells that are innervated by ascending afferent neurons (green) that collect into the posterior lateral line ganglion. Putative cholinergic efferent neurons (red) descend from the hindbrain and project onto neuromast hair cells (**A**). 2-[4-(Dimethylamino)styryl]-1-ethylpyridinium iodide (DASPEI) staining of 6 dpf surface (n = 14, **B**) and cavefish (n = 17, **C**) show significantly different quantities of anterior lateral line neuromasts (p < 0.01, **D**) and similar quantities of posterior neuromasts (p = 0.77, **E**). Hair cells were labeled with a 30-min treatment of YO-PRO1 (green) and found to be of similar quantities between surface (n = 9) and cave (n = 11) populations (unpaired two-way t-test, p = 0.09, **F**). Box-and-whisker plot representing the number of hair cells per neuromast was indistinguishable across development (5–7 dpf; p = 0.06, **G**). Error bars are ± standard error (SE) and ** denotes a significant difference (p < 0.01) detected by an unpaired t-test.

Neuromasts along the posterior lateral line were comprised of similar quantities of hair cells across blind (n = 11) and surface morphs (n = 9; p = 0.09; **Figure 1F**). No significant interaction was detected between the effects of population and age (5–7 dpf; $F_{2,72}$ = 2.96, p = 0.06; **Figure 1G**). We exclusively probed the posterior lateral line afferent neurons to establish whether sensory systems that are anatomically similar exhibit neurophysiological differences that contribute to enhanced sensitivity.

To examine the physiological basis of differences in lateral line function across surface and cavefish we used extracellular lateral line recordings adapted from protocols used in zebrafish (**Figure 2A, B**). Extracellular recordings of posterior lateral line afferents revealed intrinsic spontaneous activity was higher in Pachón cavefish (18.6 ± 0.2 Hz) while the animal was at rest, relative to surface fish (12.4 ± 0.3 Hz; p < 0.01, t = 15.97, df = 5,795; **Figure 2C**). Single neuromasts were then deflected and evoked activity was recorded from the connected afferent neuron (**Figure 2D, E**). The neuromasts

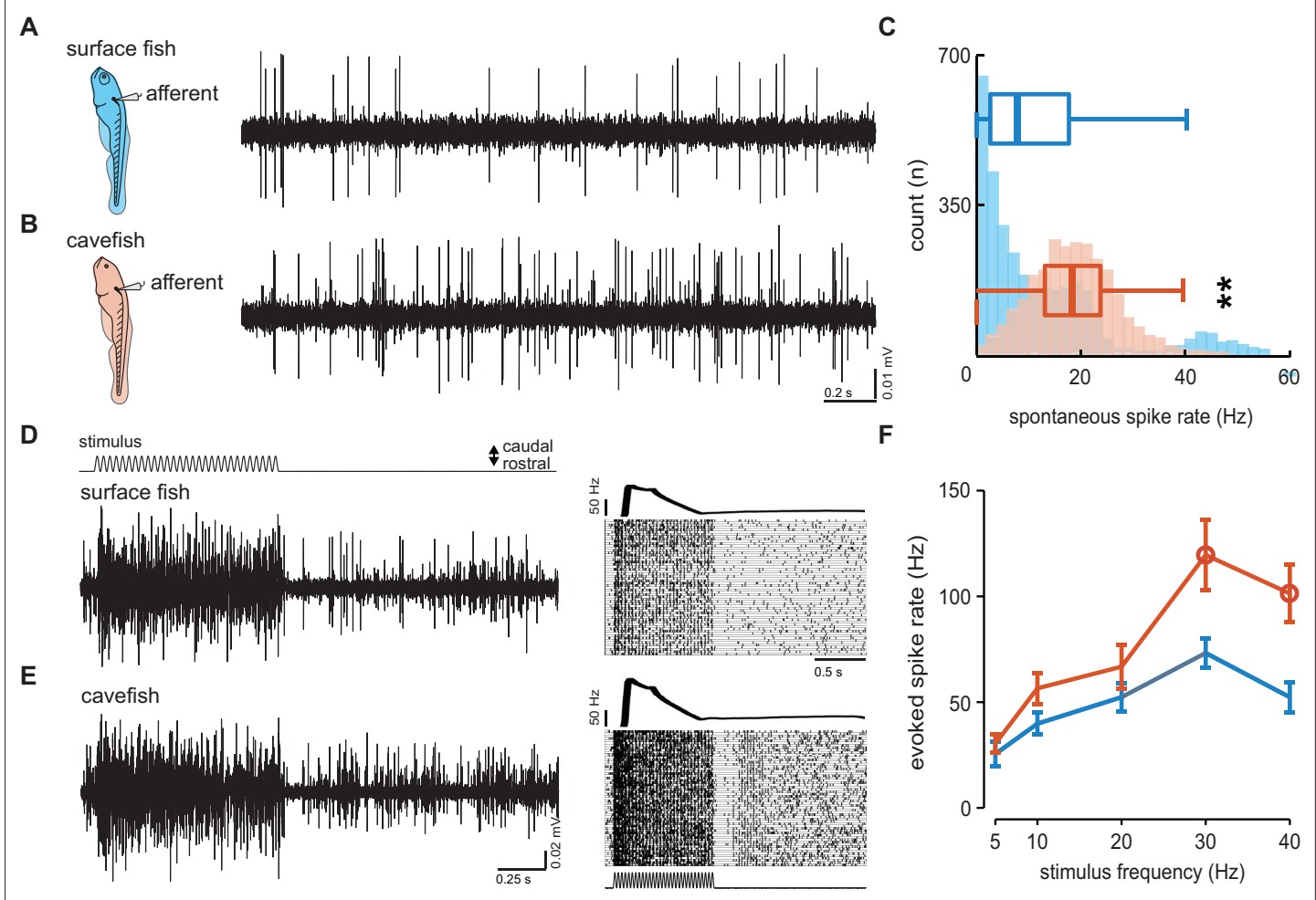

**Figure 2.** Dynamics of spontaneous and evoked afferent neuron spike activity is elevated in cavefish. Extracellular recordings were made in posterior lateral line afferents where the neuromast densities and hair cell quantities were similar to resolve the differences observed in afferent activity between larval surface fish (**A**) and Pachón cavefish (**B**) between 4 and 7 dpf. The number of occurrences and median intrinsic spike rates in both surface (blue; 12.4 Hz, *n* = 10 fish) and Pachón (red; 18.6 Hz, *n* = 5 fish) fish suggests that lateral line response thresholds in cavefish are lower than those of surface fish (**C**). Evoked afferent activity during stimulation of a single neuromast in surface fish (n = 10, **D**) and cavefish (n = 12, **E**) and peristimulus time histograms demonstrate elevated spike rate in cavefish. Evoked spike rate was stimulus frequency dependent and pairwise comparisons reveal significantly (open circles) elevated sensitivity in cavefish at 30 and 40 Hz (p = 0.03 and p < 0.01, respectively, **F**). Error bars are ± standard error (SE) and ** denotes a significant difference (p < 0.01) detected by an unpaired t-test.

were stimulated at different frequencies (5–40 Hz) and a pairwise comparison of evoked afferent activity at 30 Hz revealed statistically elevated spike rates in Pachón cavefish (*n* = 12; 119.6 ± 16.6 Hz) relative to surface fish (*n* = 10; 73.2 ± 6.9 Hz; p = 0.03, *t* = 2.38, df = 18; *Figure 2F*). Significantly elevated spike rates were also detected in response to a 40-Hz stimulus in Pachón cavefish (101.3 ± 13.6 Hz) relative to surface fish (52.3 ± 7.0 Hz; p < 0.01, *t* = 3.00, df = 18; *Figure 2F*). Together these findings support spontaneous afferent activity as a reliable metric for estimating the sensitivity of the lateral line and reveal enhanced neural responses to hair cell deflection in cavefish.

We examined afferent signaling during motor commands in our paralyzed preparation which were denoted by bouts of fictive swimming (hereafter swimming). Swim bouts of longer duration were observed in surface fish (357 ± 5 ms; *n* = 3167 swim bouts) when compared to Pachón cavefish (264 ± 4 ms; *n* = 2612 swim bouts; p < 0.01, *t*-stat = 15.2, df = 5777). Instantaneous afferent spike rate demonstrates substantial decreases during swimming in surface fish and little effect in Pachón cavefish (*Figure 3A, B*). We quantified and compared spike rates during swimming relative to the pre-swim period to examine patterns of the inhibitory effect between populations. During most surface fish swim bouts there was a reduction in afferent activity (n = 1966/2291, 85.8%), many of which resulted

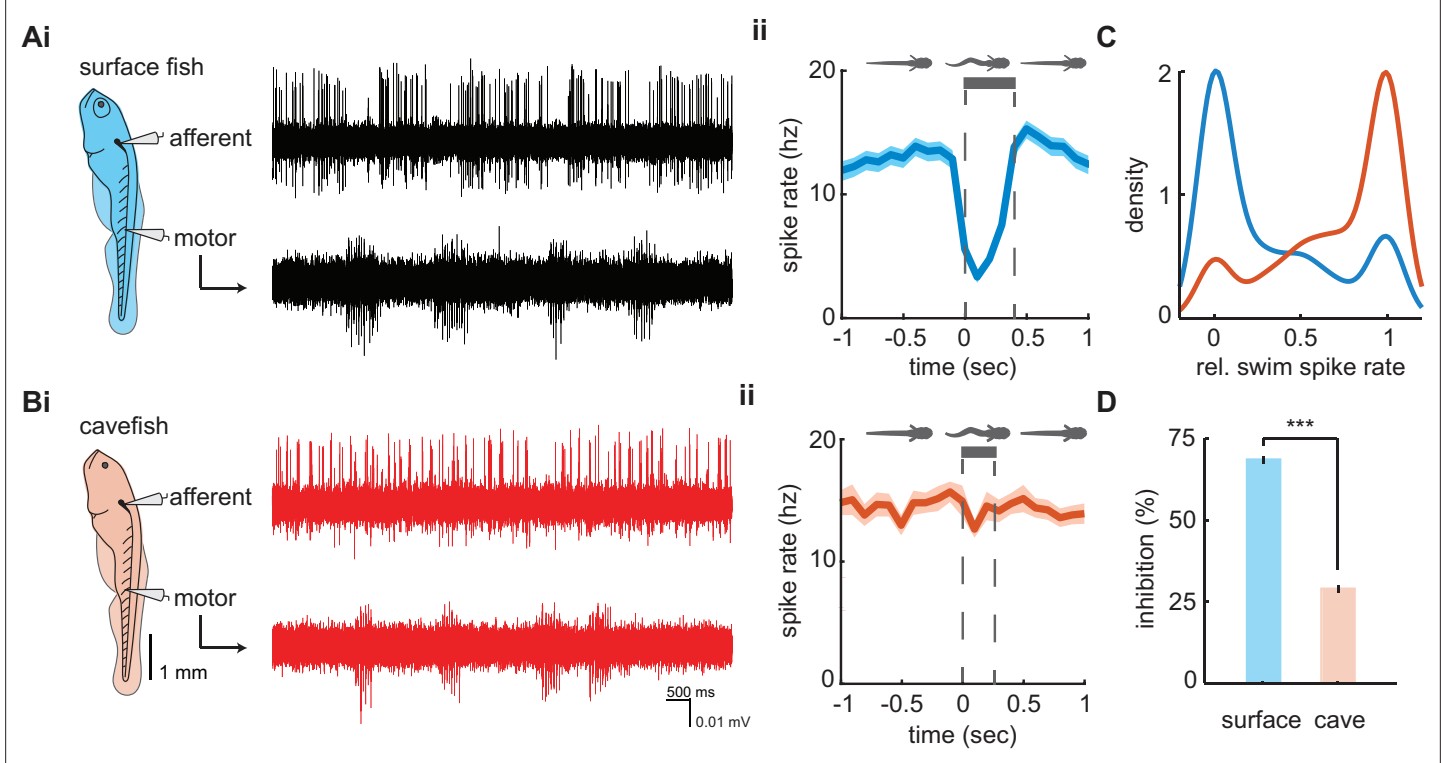

**Figure 3.** Afferent spike rate decreases during swimming in surface fish but not cavefish. Simultaneous recordings from afferent neurons from the posterior lateral line afferent ganglion and ventral motor roots along the body in (**Ai**) surface fish and (**Bi**) Pachón cavefish. Afferent spike rate decreases at the onset of swimming (time = 0) in (**Aii**) surface fish while (**Bii**) Pachón cavefish spike rate remained relatively constant during swimming. Bars represent average swim duration for surface fish (357 ± 5 ms, *n* = 2272 swims) and Pachón cavefish (264 ± 4 ms, *n* = 2612 swims). (**C**) Kernel density estimate of spike rate during swimming relative to the pre-swim interval in both surface fish (blue) and cavefish (red). (**D**) Surface fish experience greater levels of inhibition during swimming than cavefish. Significance level p < 0.001 indicated by '***' detected via unpaired two-way t-test. Error bars are ± standard error (SE).

in complete quiescence of transmissions (*n* = 1112/2291, 48.5%). Conversely, afferent activity partially reduced during many swim bouts in Pachón cavefish (*n* = 1303/2439, 53.4%), but very few instances led to complete inhibition (*n* = 275/2439, 11.3%). The distributions of relative spike rates during swimming reveal surface fish have a higher likelihood of experiencing no afferent activity during swimming while cavefish experience afferent activity during swimming similar to that of pre-swim activity levels (*Figure 3C*). Therefore, surface fish experience significantly higher levels of inhibition (68.5 ± 0.01%) compared to cavefish (28.9 ± 0.01%; p < 0.01, $t$ = 36.5, df = 4449; *Figure 3D*). Surface fish demonstrate lateral line inhibition during swimming comparable to other fishes with intact visual systems (*Flock and Russell, 1973*; *Lunsford et al., 2019*; *Pichler and Lagnado, 2020*; *Russell and Roberts, 1972*) suggesting cavefish have evolved a unique functional phenotype for sensory gain control.

We imaged hindbrain cholinergic efferent neurons to determine anatomical and functional connectivity. Between populations, backfilled efferents revealed similar soma quantities (surface: 2.4±0.3; cave: 3.2±0.5; p = 0.2, $t$ = 1.36, df = 18) and size (surface: 62.1 ± 3.6 µm$^2$; cave: 70.9 ± 4.5 µm$^2$; p = 0.1, $t$ = 1.50, df = 51; *Figure 4A–C*). From electrophysiological recordings, we observed average spike rates during and prior to a swim were not positively correlated in control surface fish ($r^2$=0.15, $F_{1,13}$ = 2.3, p = 0.2), but the slope of the line indicates a fractional suppression of 78% that is significantly less than unity (slope 0.1, confidence interval [CI]: −0.2 to 0.4; *Figure 4Di*). In the efferent-ablated surface fish, the spike rates during swimming intervals were indistinguishable from those during nonswimming intervals (slope 1.2, CI = −0.1 to 2.4) showing no detectable inhibitory effect. While swimming, surface fish without functioning efferent neurons demonstrated a signaling phenotype similar to both intact cavefish (slope 0.88, CI = −0.5 to 2.2) and cavefish with ablated efferents (slope 0.9, CI = 0.1 to 1.8; *Figure 4Dii*). Therefore, putative cholinergic hindbrain efferents are necessary for afferent inhibition in surface fish, but do not demonstrate modulatory control of afferents in cavefish.

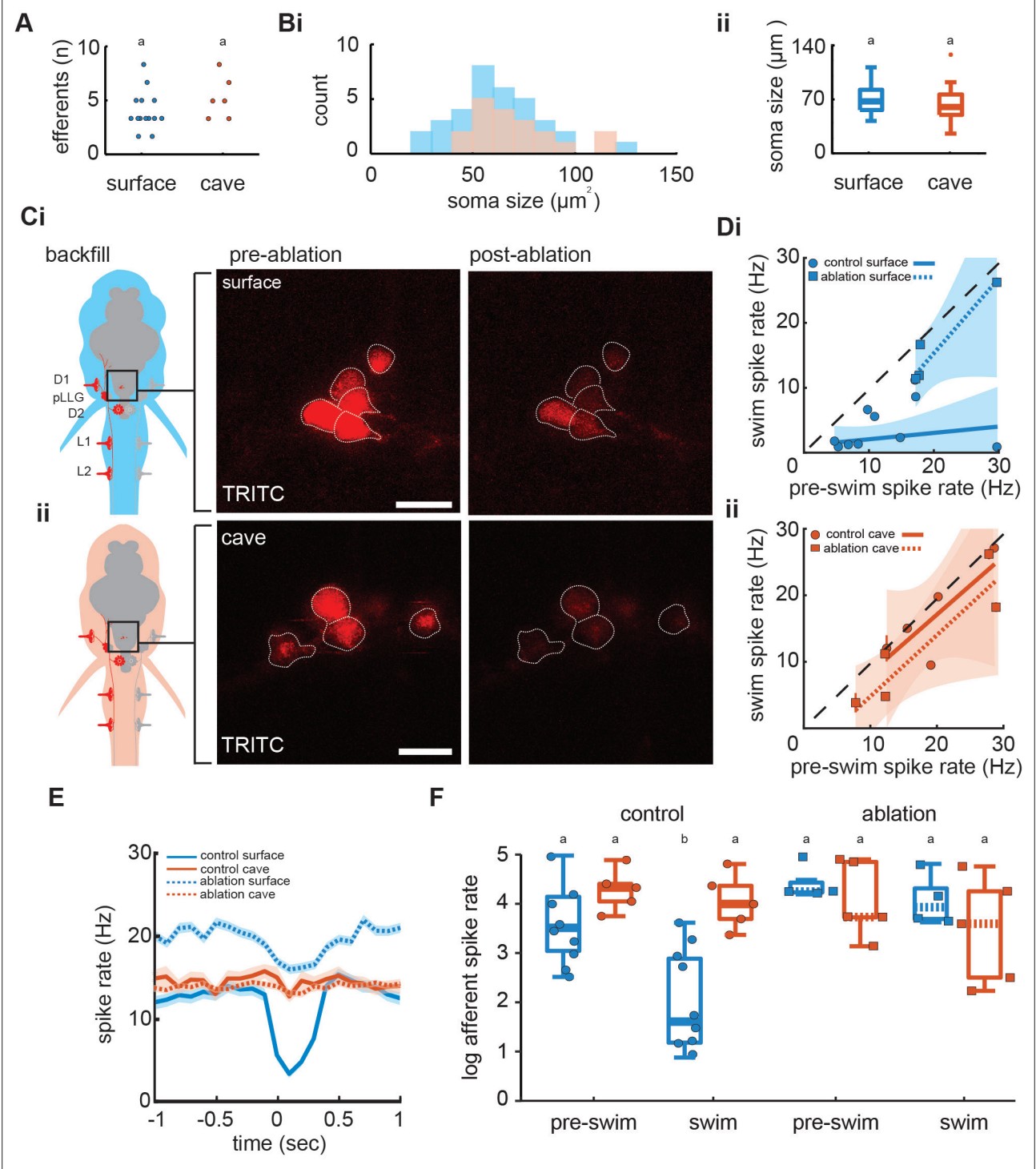

**Figure 4.** Efferent neurons are necessary for inhibition observed in afferents during swimming in surface fish but not cavefish. (**A**) Backfilled hindbrain cholinergic efferent neurons were present in comparable numbers (2–3 cells) in both surface fish (*n* = 14) and cavefish (*n* = 6, unpaired two-way t-test: p = 0.2). (**Bi**) Efferent soma size in surface (blue) and cavefish (red) is similar in both populations (unpaired two-way t-test: p = 0.1, **ii**). (**C**) Efferent cell bodies were identified by backfilling rhodamine through posterior lateral line neuromasts in both surface fish (**i**) and cavefish (**ii**) and were ablated with a 30-s UV pulse. Scale bar: 20 μm. (**D**) The line of best fit of spike rates before compared to during the swim significantly excludes unity in nonablated, control surface fish (CI: -0.2 to 0.4, circle), but not in ablated surface fish (CI: -0.1 to 2.4, square), implying spike rate suppression in the former but not the latter (**i**). The slope of the line for control fish suggests the inhibition is not correlated to the spontaneous afferent activity preceding the swim (p = 0.2). The line of best fit of Pachón cavefish pre-swim and swim spike rates did not exclude unity in both control (CI: -0.5 to 2.2, circle) and ablated (CI: 0.9 to 1.8, square) treatments (**ii**). Dashed line indicates the line of unity, corresponding to no average difference of spike rate during swimming. (**E**)

*Figure 4 continued on next page*

*Figure 4 continued*

Instantaneous spike rates of Pachón cavefish were unaffected by ablating the lateral line whereas the inhibitory effect was eliminated in ablated surface fish. Time is relative to the onset of motor activity. (**F**) Surface fish (blue) display reduced spike rates during swimming compared to before swimming in control fish (Tukey's post-hoc test: p < 0.01). Pachón cavefish (red) did not display reduced spike rate during swimming in neither control nor ablated treatments. Ablated surface individuals also did not display reduced spike rate during swimming resulting in a signaling phenotype comparable to cavefish. Groupings of statistical similarity are denoted by 'a' and 'b', whereas a is significantly different from b. All error bars represent ± standard error (SE).

We compared pre-swim and swim spike rates across populations and treatments to determine efferent contribution to inhibition (*Figure 4E*). We found significant differences in afferent activity among pre-swim and swim intervals ($F_{7,40}$ = 7.6, p < 0.01; *Figure 4F*). Surface fish afferent spike rates during swimming (3.9 ± 0.1 Hz) were 71% lower than the immediate pre-swim period in control fish (13.7 ± 0.2 Hz; Tukey's post hoc test, p < 0.01). Post-swim spike rate (13. 7 ± 0.3 Hz) recovered to pre-swim spike rate. In control Pachón cavefish, we observed some decrease in afferent spike rate during swimming (16.8 ± 0.2 Hz) but it was not statistically discriminated from pre-swim (20.5 ± 0.2 Hz; Tukey's post hoc test, p = 0.94). Efferent ablation in surface fish resulted in afferent activity during swimming (17.4 ± 0.3 Hz) to increase to pre-swim levels (20.7 ± 0.3 Hz; Tukey's post hoc test, p = 0.99). Ablated surface fish also demonstrated spike dynamics during swimming comparable to ablated and control cavefish (Pachón ablated, pre-swim: 18.6 ± 0.3 Hz; Pachón ablated, swimming: 14.7 ± 0.3 Hz). These findings indicate that efferents are necessary for inhibition of afferents during swimming in surface fish.

We compared lateral line activity between three different cave populations (*Figure 5A*); the Tinaja and Pachón populations, which are derived from similarly timed colonization events, and the Molino population that is derived from a more recent colonization event (*Bradic et al., 2012*; *Dowling et al., 2002*; *Herman et al., 2018*). Blind cavefish populations exhibited similar spontaneous spike rates across populations ($F_{2,17}$ = 0.68, p = 0.52), and swimming showed little effect on lateral line activity across blind cavefish populations (*Figure 5B*). We observed a minor decrease in afferent spike rates across cavefish populations and the relative change (i.e., inhibition) was similar in Pachón and Tinaja (Tukey's post hoc test; Pachón: 0.28 ± 0.01; Tinaja: 0.32 ± 0.02; *Figure 5C*). Molino demonstrated an intermediate phenotype compared to Pachón and Tinaja (Tukey's post hoc test; Molino: 0.23 ± 0.01; p < 0.01). The Molino population (new lineage) is most distantly related to the Pachón and Tinaja populations (old lineage), originating from a more recent surface fish colonization of caves thus providing phylogenetic evidence that coincides with statistically similar groupings (*Herman et al., 2018*). We examined the correlation between nonswimming and swimming spike rates across cave populations to determine whether the inhibitory effect was significant. Spike rates during swimming and pre-swim intervals were positively correlated in all cave populations, and afferent spike rates during swimming were not statistically distinguishable from unity across populations (Pachón: slope = 0.88, CI = −0.5 to 2.2; Tinaja: slope = 1.03, CI = −0.2 to 2.2; Molino: slope = 1.02, CI = 0.8 to 1.2; *Figure 5D*), indicating there was no significant decrease in afferent activity during swimming within blind cavefish populations.

## Discussion

Our principal findings indicate elevated afferent spike activity and an increased responsiveness to hair cell deflection in cavefish. The increased afferent activity in cavefish is a likely consequence of eye loss, which has robust effects on other physiological systems (*Duboué et al., 2011*; *Varatharasan et al., 2009*). Heightened lateral line sensitivity in adults has been previously attributed to increased neuromast density in the ALL as well as increased hair cell quantities per neuromast (*Jaggard et al., 2017*; *Lloyd et al., 2018*; *McHenry et al., 2008*; *Patton et al., 2010*; *Teyke, 1990*; *Yoffe et al., 2020*; *Yoffe et al., 2020*; *Yoshizawa et al., 2010*). These differences do not exist in the posterior lateral line at the larval stage, allowing us a unique opportunity to investigate the neural mechanisms that can enhance flow sensing in a strong model for evolution. Our discovery of increased spontaneous activity that contributes to the enhanced evoked responses in cavefish is a powerful addition for flow sensing, and we predict that this in combination with the increased number of neuromasts in the ALL (*Yoshizawa et al., 2010*) is what ultimately enables cavefish to perform active-flow sensing.

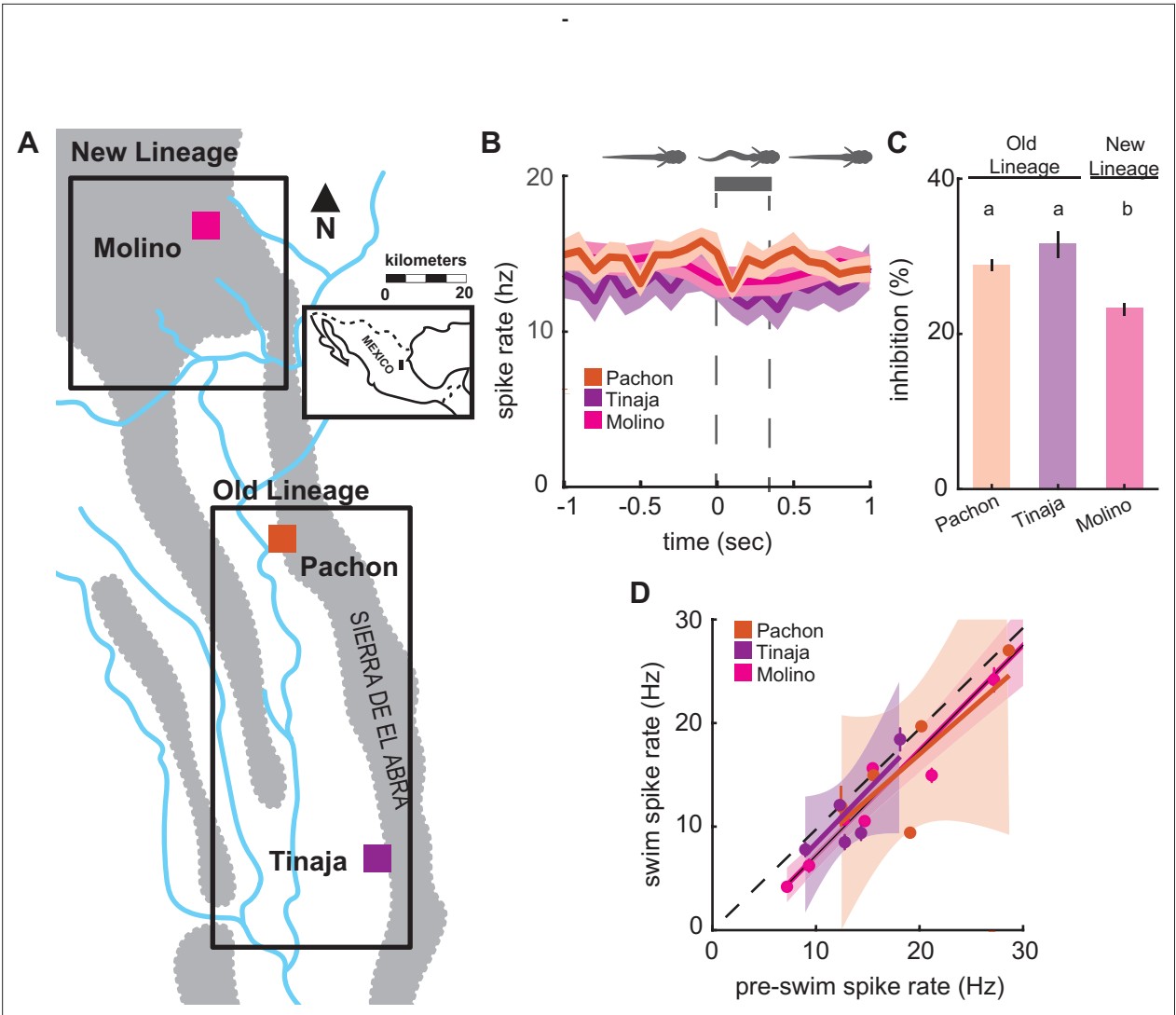

**Figure 5.** Enhanced lateral line sensitivity during swimming convergently evolved across three blind populations. (**A**) Molino cave populations (pink; New Lineage) have evolved more recently relative to Pachón (red) and Tinaja (purple; Old Lineage) cave populations. Lineage delineations inferred from phylogenetic data (*Herman et al., 2018*). (**B**) Mean spontaneous afferent spike rate remains constant at the onset of fictive swimming (time = 0) in Pachón (*n* = 5), Molino (*n* = 8), and Tinaja (*n* = 5) populations (N-way ANOVA: p = 0.52).Bar represents average swim duration for Pachón (0.27 s, *n* = 2429 swim bouts), Molino (0.42 s, *n* = 1,474 swim bouts), and Tinaja (0.35 s, *n* = 464). (**C**) Percent change in spike rate from pre-swim to swim intervals (i.e., inhibition) was small, but significantly different between blind cave populations. Post hoc Tukey test revealed that Molino cavefish experienced significantly less reduction in spike rate when compared to Pachón and Tinaja populations (p < 0.01). Statistically similar groups are indicated by 'a' and 'b'. (**D**) The line of best fit of pre-swim and swim spike rates does not significantly exclude unity in any of the blind cavefish populations implying there is no detectable inhibitory effect (Pachón: CI = -0.5 to 2.2, Tinaja: CI = -0.2 to 2.2, Molino: CI = 0.8 to 1.2). Dashed line indicates the line of unity, corresponding to no average difference of spike rate during swimming. All values represent mean ± standard error (SE).

Higher spontaneous activity is one of two mechanisms that are responsible for higher lateral line sensitivity. The other involves the inhibitory effects of the efferent system during swimming, a feedback mechanism that is conserved across the diversity of fishes (*Flock and Russell, 1973*; *Lunsford et al., 2019*; *Montgomery et al., 1996*; *Montgomery and Bodznick, 1994*; *Roberts and Russell, 1972*; *Tricas and Highstein, 1991*). This is true in swimming surface fish but not in cavefish. We found that three blind populations of cavefish (i.e., Pachón, Molino, and Tinaja) have repeatedly lost the capability for efferent modulation during swimming. When one considers that lateral line efferent activity can be driven by direct inputs from the visual system in zebrafish (*Reinig et al., 2017*), it seems possible that eye degeneration induces the loss of efferent function. This interpretation is consistent with the idea that lateral line efferents are thought to have undergone regressive loss before in the

ancestral lamprey and hagfish (*Kishida et al., 1987*; *Köppl, 2011*; *Koyama et al., 1990*), both of which are nearly or completely blind during development (*Dickson and Collard, 1979*; *Fernholm and Holmberg, 1975*). However, our results illustrate that cholinergic efferent system is still present in cavefish, having lost functionality rather than disappearing (the efferent system is functional in surface fish). Exploring pre- and postsynaptic differences such as acetylcholine release or the density of nicotinic acetylcholine receptors (nAChR) may explain the reduced inhibitory efficacy and reveal molecular targets that could disrupt efferent function over the course of evolution (*Dawkins et al., 2005*).

Our demonstration of CD inactivity in cavefish provides an alternative mechanism by which evolution can enhance sensitivity, one that proceeds by decreasing inhibition rather than augmenting sensor morphology or density (*Yamamoto et al., 2009*; *Yoshizawa et al., 2010*; *Yoshizawa et al., 2014*). The impact of increasing sensitivity through a lack of inhibition is apparent during active-flow sensing in adult *A. mexicanus*. Active-flow sensing occurs during swimming and involves using the reflection of self-generated flow fields (*Bleckmann et al., 1991*) to follow walls (*Patton et al., 2010*), avoid obstacles (*Teyke, 1985*; *Windsor et al., 2008*), and discrimination between shapes (*de Perera, 2004*; *Hassan, 1989*; *von Campenhausen et al., 1981*). The repeated loss of inhibition across populations at the larval stage suggests that selection may act early in development on the efferent over afferent systems. Future experiments may test this hypothesis in adults, though it seems likely that this phenomenon would be preserved and only elaborated on throughout ontogeny, rather than reconfigured. Cavefish swim by using different body motions than surface fish. This finding has been interpreted as a mechanism to enhance wall following, which occurs when the bow wake of a swimming cavefish is reflected off of a solid surface and then detected (*Patton et al., 2010*; *Sharma et al., 2009*). Altered swimming kinematics are also thought to have arisen from a general increase in sensitivity to flow (*Tan et al., 2011*; *Windsor et al., 2008*). Our results provide an alternate suggestion; the lack of efferent function in cavefish precludes the sensory feedback necessary for sensing self-movement and body position in water (proprioception). CD, a parallel motor command that decreases the afferent activity of fishes when swimming, has recently been found to play a critical role in swimming efficiency by enabling the tracking of the traveling body wave during undulation (*Skandalis et al., 2021*). We hypothesize that the evolved loss of efferent function that enables cavefish to successfully avoid collisions in subterranean habitats is likely favored over optimizing swimming efficiency (*Nakamura, 1997*; *Atasavun Uysal et al., 2010*). We predict that loss of efferent function will be found in other blind hypogean species (*Costa Sampaio et al., 2012*) and that their respective surface populations will possess intact efferent functionality, as we have found in *A. mexicanus*. Neurophysiological recordings across a wider diversity of species would provide valuable insight into how efferent function may be sculpted by environmental selection and phylogenetic membership.

By employing neurophysiological approaches in the lateral line system of *A. mexicanus* for the first time, we show that elevated lateral line afferent activity and loss of efferent function have repeatedly evolved together across cavefish populations. Our findings come at a time when genetic tools in *A. mexicanus* enable brain-wide imaging and gene-editing based screening to identify candidate neural circuits and genes critical in the evolution of sensory systems (*Jaggard et al., 2020*; *Warren et al., 2021*). Going forward, applying genetic and electrophysiological tools in well-characterized neural circuits promises to inform our understanding of the evolution of neural systems and behavior more broadly.

## Materials and methods
### Animals
Fish were progeny of pure-bred stocks originally collected in Mexico (*Duboué et al., 2011*) that have been maintained at the Florida Atlantic University core facilities. Larvae were raised in 10% Hank's solution (137 mM NaCl, 5.4 mM KCl, 0.25 mM $Na_2HPO_4$, 0.44 mM $KH_2PO_4$, 1.3 mM $CaCl_2$, 1.0 mM $MgSO_4$, 4.2 mM $NaHCO_3$; pH 7.3) at 26°C. All experiments were performed according to protocols approved by the University of Florida or Florida Atlantic University Institutional Animal Care and Use Committee (IACUC201603267, IACUC202200000056). Animal health was assessed by monitoring blood flow throughout each experiment.

## Neuromast imaging

To assess neuromast quantities, larvae aged 6 dpf were submerged in 5 µg/ml DASPEI dissolved in embryo medium for 15 min as previously described (*Van Trump et al., 2010*). Larvae were then transferred to ice-cold water for 30–45 s then immersed in 8% methylcellulose for imaging. Images were taken using a Nikon DS-Qi2 monochrome microscope camera mounted on a Nikon SMZ25 Stereo microscope (Nikon; Tokyo, Japan). Neuromasts innervated by posterior lateral line afferents and ALL afferents were tabulated separately. To image hair cells that comprise each neuromast, larvae (5–7 dpf) were immersed in 2 µM YO-PRO1 (Invitrogen; Y3603) in embryo media for 30 min, rinsed three times in embryo media (*Santos et al., 2006*). Larvae were then embedded on their sides in low-melting point-1.6% agarose and imaged on a confocal microscope (Leica TCS SP5, ×63/1.2 water immersion, 200 Hz, emission: 644–698 nm). Images were processed on ImageJ (v1.48; U. S. National Institutes of Health, Bethesda, MD).

## Electrophysiology

Prior to recordings, *A. mexicanus* larvae (4–7 dpf) were paralyzed using 0.1% α-bungarotoxin (*Lunsford and Liao, 2021*). Once paralyzed, larvae were then transferred into extracellular solution (134 mM NaCl, 2.9 mM KCl, 1.2 mM $MgCl_2$, 2.1 mM $CaCl_2$, 10 mM glucose, 10 mM HEPES buffer; pH 7.8, adjusted with NaOH) and pinned with etched tungsten pins through their dorsal notochord and otic vesicle into a Sylgard-bottom dish.

Multiunit extracellular recordings of the posterior lateral line afferent ganglion were made in surface fish ($n$ = 10) and Pachón cavefish ($n$ = 5). Prior to recording from the afferent neurons, a bore pipette was used to break through the skin to expose the afferent soma. Recording electrodes (~30 µm tip diameter) were pulled from borosilicate glass (model G150F-3, inner diameter: 0.86, outer diameter: 1.50; Warner Instruments, Hamden, CT) on a model P-97 Flaming/Brown micropipette puller (Sutter Instruments, Novato, CA) and filled with extracellular solution. Once contact with afferent somata was achieved, gentle negative pressure was applied (20–50 mm Hg; pneumatic transducer tester, model DPM1B, Fluke Biomedical Instruments, Everett, WA). Pressure was adjusted to atmospheric (0 mm Hg) once a stable recording was achieved. Simultaneously, ventral root recordings were made through the skin (*Masino and Fetcho, 2005*) to detect voluntary fictive swimming. To record response properties from afferent neurons during neuromast deflection we systematically probed neuromasts starting at D1 and continued down the body until stereotypical evoked potentials were detected (*Levi et al., 2015*). The sinusoidal stimulus was driven using a single-axis piezo stimulator (30V300, Piezosystem Jena, Hopedale, MA). The stimulus probe was formed by melting the tip of a small diameter (~2 µm) recording electrode on a MF-830 microforge (Narishige International, Amityville, NY) and the probe tip was positioned approximately 50 µm anterior to the cupula at kinocilia-tip height. Stimulation occurred at 5, 10, 20, 30, and 40 Hz which encompasses frequency ranges at the lower limit of tail beats (5–10 Hz) and typical range of tail beat during free swimming (20–40 Hz) (*Mirat et al., 2013*). The stimulus period lasted for 1 s with 4 s at rest to allow for recovery and was repeated for each frequency for 60 sweeps. Care was taken to avoid any contact between the probe and the cupula which can alter the response (*Levi et al., 2015*). All recordings were sampled at 20 kHz and amplified with a gain of 1000 in Axoclamp 700B, digitized with Digidata 1440A and saved in pClamp10 (Molecular Devices).

All recordings were analyzed in Matlab (vR2019b) using custom written scripts. Both spontaneous afferent spikes and swimming motor activity identified using a combination of spike parameters previously described (*Lunsford et al., 2019*). Afferent neuron activity within a time interval equal to the subsequent fictive swim bout, hereafter termed 'pre-swim', was quantified and compared to afferent activity during swimming to measure relative changes in spontaneous firing.

## Efferent ablations

Hindbrain efferent neurons were backfilled with tetramethylrhodamine (TRITC, 3 kDa; Molecular Probes, Eugene, OR). *A. mexicanus* larvae (4 dpf) were anesthetized in MS-222 (Tricaine, Western Chemical Inc, Ferndale, WA) and embedded in agar. To selectively label the hindbrain cholinergic efferent neurons, we systematically electroporated (Axoporator 800A Single Cell Electroporator, Axon CNS Systems, Molecular Devices LLC, San Jose, CA) TRITC into the efferent terminals that innervate the D1, D2, L1, and L2 neuromasts of the lateral line in surface fish ($n$ = 14) and Pachón cavefish ($n$ =

7; *Figure 4Ci*). Electroporation does not ensure labeling of all efferent neurons so we standardized parameters (30 V, 50 Hz, 500 ms, square pulse) and targeted the same neuromasts across populations to minimize variation in labeling efficacy. Larvae were then gently freed from the agar and allowed to swim freely and recover overnight. Larvae (5 dpf) were then paralyzed via α-bungarotoxin immersion, remounted in agar dorsal surface down, and imaged on a Leica SP5 confocal microscope (Leica Microsystems, Wetzlar, Germany). Efferent soma size and quantity were measured within identified TRITC-labeled cells in ImageJ (v1.48; U. S. National Institutes of Health, Bethesda, MD). To perform targeted ablations of surface fish (*n* = 5) and cavefish (*n* = 6) efferent neurons, a near-ultraviolet laser was focused at a depth corresponding to the maximum fluorescence intensity of each soma, to ensure we were targeting its center. We applied the FRAP Wizard tool in Leica application software to target individual cells. We ablated target cells with a 30-s exposure to the near-ultraviolet laser line (458 nm), and successful targeting was confirmed by quenching of the backfilled dye. This method has been successfully applied and validated in similar systems (*Liu and Fetcho, 1999*; *Soustelle et al., 2008*). Fish were again freed from agar and allowed to swim freely and recover overnight. Electrophysiological recordings were performed on ablated surface fish (*n* = 4) and cavefish (*n* = 5) at 6 dpf to simultaneously monitor afferent activity and motor activity.

## Statistical analysis

Neuromast data were analyzed using GraphPad Prism 8.4.3. Normality was assessed via Shapiro–Wilk test. ALL neuromast count data were found to be normally distributed. ALL neuromast quantities in surface and cave larvae were compared using an unpaired *t*-test. Posterior lateral line neuromast count was found to not be normally distributed and was subsequently analyzed via Mann–Whitney *U*-test neuromast hair cell data were analyzed using MatLab (v2019b). Surface and cavefish hair cell counts per neuromast compared using an unpaired two-way *t*-test.

Analyses of electrophysiological data (*Supplementary file 1*) were performed using custom written models in the R language (R development core team, vR2016b) using packages car, visreg, reshape2, plyr, dplyr, ggplot2, gridExtra, minpack.lm, nlstools, investr, and cowplot (*Auguie et al., 2017*; *Baty et al., 2015*; *Breheny and Burchett, 2017*; *Fox and Weisberg, 2018*; *Wickham, 2007*; *Wickham, 2011*; *Wickham et al., 2019*; *Wilke, 2019*) and Matlab (v2019b). Spontaneous afferent spike rate was calculated by taking the number of spikes over the duration of time where the larva was inactive and the neuromast was not being stimulated. Instantaneous afferent spike rate was calculated using a moving average filter and a 100ms sampling window. Evoked afferent spike rate was calculated by taking the number of spikes over the duration of the stimulus period (1 s). Pre-swim and swim spike rate were calculated by taking the number of spikes within the respective period over its duration. Pre-swim periods of inactivity made it challenging to interpret changes in afferent activity, so we restricted the dataset to only include swim bouts that were preceded by a minimum of one afferent spike within the pre-swim interval (surface = 2291 swim bouts; Pachón = 2429 swim bouts; Molino = 1.474 swim bouts; Tinaja = 464 swim bouts). Swim frequency was calculated by taking the number of bursts within a swim bout over the duration of the swim bout. Relative spike rate was calculated by taking the swim spike rate over the pre-swim spike rate. All variables were averaged for each individual fish. The precision of estimates for each individual is a function of the number of swims, so we analyzed variable relationships using weighted regressions, with individual weights equal to the square root of the number of swims. We log transformed variables in which the mean and the variance were correlated. To quantify the inhibition of the afferent spike frequency during swimming we tested for a significant difference in afferent spike frequency during swimming as compared to nonswimming periods using a paired sample Student's *t*-test.

Differences in afferent spike rates between the periods of interest (pre-swim and swim) across populations and treatments were tested by *N*-way analysis of variance followed by Tukey's post hoc test to detect significant differences in spike rates between swim periods or treatments. Linear models were used to detect relationships between spike rate during swimming and other independent variables (e.g., spike rate pre-swim). Data are shown throughout the manuscript as mean ± standard error. Statistical significance is reported at $\alpha$ = 0.05.

## Acknowledgements

We gratefully acknowledge support from the US-Israel BSF SP#2018-190, National Science Foundation (IOS165674), and National Institute of Health (1R01GM127872) to ACK, and the National Institute of Health (DC010809), National Science Foundation (IOS1856237, IOS2102891), and support from the Whitney Laboratory for Marine Biosciences to JCL.

## Additional information

### Funding

| Funder | Grant reference number | Author |
| --- | --- | --- |
| US-Israel Binational Science Foundation | SP#2018-190 | Alex C Keene |
| National Science Foundation | IOS165674 | Alex C Keene |
| National Institutes of Health | IR01GM127872 | Alex C Keene |
| National Institutes of Health | DC010809 | James C Liao |
| National Science Foundation | IOS1856237 | James C Liao |
| National Science Foundation | IOS2102891 | James C Liao |

The funders had no role in study design, data collection, and interpretation, or the decision to submit the work for publication.

### Author contributions

Elias T Lunsford, Conceptualization, Data curation, Formal analysis, Investigation, Methodology, Visualization, Writing - original draft, Writing – review and editing; Alexandra Paz, Formal analysis, Investigation, Methodology, Writing – review and editing; Alex C Keene, Funding acquisition, Resources, Writing – review and editing; James C Liao, Conceptualization, Funding acquisition, Methodology, Project administration, Resources, Supervision, Validation, Writing – review and editing

### Author ORCIDs

Elias T Lunsford (iD) http://orcid.org/0000-0002-3713-6994
James C Liao (iD) http://orcid.org/0000-0003-0181-6995

### Ethics

All animals were handled according to protocols approved by the University of Florida or Florida Atlantic University Institutional Animal Care and Use Committee (IACUC201603267, IACUC202200000056). Animal health was assessed by monitoring blood flow throughout each experiment.

### Decision letter and Author response

Decision letter https://doi.org/10.7554/eLife.77387.sa1
Author response https://doi.org/10.7554/eLife.77387.sa2

## Additional files

### Supplementary files

• Supplementary file 1. Data collected from electrophysiology recordings and hair cell imaging. Afferent spike rate during motor inactivity, fictive swimming, and neuromast deflection. Data include information related to the parameters of fictive swim bouts (i.e., tail-beat frequency, swim duration, and duty cycle) as well as stimulus frequency associated with evoked activity. Imaging data denote the population (Pachón or surface), age, and neuromast identity associated with each hair cell count.

• MDAR checklist

## Data availability

Electrophysiology data generated and analyzed during this study are included as a supplementary file ('Supplementary File 1').

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
