## [Editor Report]

While most cavefish lack vision, their blindness is compensated by enhanced sensitivity to water flows. Through a powerful comparative analysis of neural circuits, the present work sheds light on how sensory mechanisms have convergently evolved in multiple populations of fish that have independently colonized different cave environments.

---

## [Decision Letter]

**Decision letter after peer review:**

Thank you for submitting your article "Evolutionary convergence of a neural mechanism in the cavefish lateral line system" for consideration by *eLife*. Your article has been reviewed by 3 peer reviewers, and the evaluation has been overseen by a Reviewing Editor and Marianne Bronner as the Senior Editor. The following individuals involved in the review of your submission have agreed to reveal their identity: Hernán López-Schier (Reviewer #1); Masato Yoshizawa (Reviewer #2); Jason Gallant (Reviewer #3).

Essential revisions:

1. A weakness of the work is that the relationship between sensitivity and spontaneous activities in the afferents was not established experimentally. The authors should tackle this weakness to support their assertion that "elevated afferent activity underlies the increased responsiveness of cavefish to flow stimuli". Neurophysiological evidence should be added to establish the increase in hair cell sensitivity. Does a sensitivity gain truly result from higher spontaneous activity? The authors should also examine potential changes in the number of mechanoreceptive hair cells.

2. Further evidence is needed to state that "elevated endogenous afferent activity identified a lower response threshold in the lateral line of blind cavefish relative to surface fish." To do this, the authors could for instance puff the trunk of the lateral line system and measure the threshold of the afferents.

*Reviewer #1 (Recommendations for the authors):*

Line 200 "Our principle finding", should be "Our principal finding".

"elevated afferent activity underlies the increased responsiveness of cavefish to flow stimuli." There is an element of wishful thinking here unless we know what happens with the hair cells.

"The repeated loss of inhibitory feedback across blind cavefish populations suggests that it is easier to cease function than to develop more neuromasts or other additive alternatives." This may be true if the fish will remain in a 6dpf stage all their lives. If you disagree, please explain.

*Reviewer #2 (Recommendations for the authors):*

L79 – 82: "Spontaneous action potential in afferent (noise activities) is essential for maintaining the state of responsiveness/sensitivity".

It is ambiguous if the increase of spontaneous spikes supports the sensitivity.

L104-105: "we identify a neurophysiological mechanism … to increase hair cell sensitivity" Does the data author presented here support this conclusion? I agree that the spontaneous afferent activities are increased, but the 'increase hair sensitivity' needs to be proved by a neurophysiological method (eg. puffing to the trunk LLS and measuring the threshold of the afferent). I believe the S/N ratio in the afferent spikes (spikes evoked by actual cupula bending as the Signal; the Noise is the spontaneous skipes of the afferent) is more biologically important for the sensitivity increase. Authors should provide the evidence for the increased sensitivity to keep their conclusion.

This evidence is also needed to support the statement in the abstract: L43: "Elevated endogenous afferent activity identified a lower response threshold in the lateral line of blind cavefish relative to surface fish." I may have missed finding their evidence, but I could not find the one that supports this statement, particularly for 'lower response threshold'.

If the authors did not show the above evidence, their abstract summary statement "… regression of the efferent system can be an evolutionary mechanism for neural adaptation of a vertebrate sensory system." is not supported.

*Reviewer #3 (Recommendations for the authors):*

I have very little to add to my overall recommendations above. I would like to provide only more detail in regards to these comments:

– I would recommend including a figure outlining the canonical circuit for lateral line function, or, at the very least better labels in figure 3Ci.

---

## [Author Response]

Essential revisions:Reviewer #1 (Recommendations for the authors):Line 200 "Our principle finding", should be "Our principal finding".

Edit is now incorporated

"elevated afferent activity underlies the increased responsiveness of cavefish to flow stimuli." There is an element of wishful thinking here unless we know what happens with the hair cells.

We have edited this sentence to now incorporate our new electrophysiology results.

“Our principal findings indicate elevated endogenous afferent spike activity that suggests an increased responsiveness to hair cell deflection in cavefish.”

"The repeated loss of inhibitory feedback across blind cavefish populations suggests that it is easier to cease function than to develop more neuromasts or other additive alternatives." This may be true if the fish will remain in a 6dpf stage all their lives. If you disagree, please explain.

The reviewer raises a fair point and we have edited the sentence accordingly.

Reviewer #2 (Recommendations for the authors):L79 – 82: "Spontaneous action potential in afferent (noise activities) is essential for maintaining the state of responsiveness/sensitivity".

We thank the reviewer for bringing this to our attention. We have revised accordingly in light of these comments.

It is ambiguous if the increase of spontaneous spikes supports the sensitivity.L104-105: "we identify a neurophysiological mechanism … to increase hair cell sensitivity" Does the data author presented here support this conclusion? I agree that the spontaneous afferent activities are increased, but the 'increase hair sensitivity' needs to be proved by a neurophysiological method (eg. puffing to the trunk LLS and measuring the threshold of the afferent). I believe the S/N ratio in the afferent spikes (spikes evoked by actual cupula bending as the Signal; the Noise is the spontaneous skipes of the afferent) is more biologically important for the sensitivity increase. Authors should provide the evidence for the increased sensitivity to keep their conclusion.This evidence is also needed to support the statement in the abstract: L43: "Elevated endogenous afferent activity identified a lower response threshold in the lateral line of blind cavefish relative to surface fish." I may have missed finding their evidence, but I could not find the one that supports this statement, particularly for 'lower response threshold'.If the authors did not show the above evidence, their abstract summary statement "… regression of the efferent system can be an evolutionary mechanism for neural adaptation of a vertebrate sensory system." is not supported.

Reviewer 2 raises an important point, to provide further evidence in support of our conclusion we performed an iteration of Reviewer 2’s suggested experimental approach. Briefly, we performed additional loose-patch recordings of posterior lateral line afferent neurons as described in the initial submission and systematically deflected each neuromast along the trunk of the larvae to identify the connected neuromast. We then provided targeted stimulation to the neuromast across a sweep of frequencies using a single axis piezo stimulator. The evoked potentials were then recorded from the corresponding afferent neurons. From these experiments, a stimulus frequency dependent response was generated revealing that cavefish are more responsive at higher stimulus frequencies than surface fish (Figure 2 F).

Reviewer #3 (Recommendations for the authors):I have very little to add to my overall recommendations above. I would like to provide only more detail in regards to these comments:– I would recommend including a figure outlining the canonical circuit for lateral line function, or, at the very least better labels in figure 3Ci.

We have added an additional panel to Figure 1 that is a simplified illustration of the relevant canonical lateral line circuit to provide readers with the essential context. We appreciate the reviewer’s recommendation.